# Brain-derived endothelial cells are neuroprotective in a chronic cerebral hypoperfusion mouse model

Yuichi Matsui[1,2,6], Fumitaka Muramatsu[1,6], Hajime Nakamura [2], Yoshimi Noda[1], Kinnosuke Matsumoto[1], Haruhiko Kishima [2] & Nobuyuki Takakura [1,3,4,5] ✉

Whether organ-specific regeneration is induced by organ-specific endothelial cells (ECs) remains unelucidated. The formation of white matter lesions due to chronic cerebral hypoperfusion causes cognitive decline, depression, motor dysfunction, and even acute ischemic stroke. Vascular ECs are an important target for treating chronic cerebral hypoperfusion. Brain-derived ECs transplanted into a mouse chronic cerebral hypoperfusion model showed excellent angiogenic potential. They were also associated with reducing both white matter lesions and brain dysfunction possibly due to the high expression of neuroprotective humoral factors. The in vitro coculture of brain cells with ECs from several diverse organs suggested the function of brain-derived endothelium is affected within a brain environment due to netrin-1 and Unc 5B systems. We found brain CD157-positive ECs were more proliferative and beneficial in a mouse model of chronic cerebral hypoperfusion than CD157-negative ECs upon inoculation. We propose novel methods to improve the symptoms of chronic cerebral hypoperfusion using CD157-positive ECs.

Although the brain makes up only 2% of total body weight, its blood flow is 20% of cardiac output[1] and contributes significantly to the maintenance of vital activity. A slow and sustained decrease in cerebral blood flow is termed chronic cerebral hypoperfusion. Such cerebral hypoperfusion disrupts the blood–brain barrier (BBB), and activates molecular and cytotoxic cascades that induce neurodegeneration and contribute to white matter damage[2,3]. White matter lesions are associated not only with cognitive decline but also with depression and motor dysfunction, and may even lead to acute ischemic stroke[4]. White matter damage is one of the risks involved with an aging population in our society[2]. Although improving chronic cerebral hypoperfusion can reduce these risks, an established treatment does not exist in clinical practice, and nonspecific treatment focuses on the management of risk factors such as the prevention and treatment of hypertension and other lifestyle-related diseases. Various therapeutic approaches exist in studies reported to date, including pharmacological approaches such as the use of minocycline, cilostazol, and edaravone[5–7]. However, rehabilitation approaches, such as brain reperfusion rehabilitation therapy[8], and

transplantation approaches, such as human umbilical mesenchymal stem cell transplantation[9] therapy, are not widely used in clinical practice.

The interaction between cells of the nervous and vascular systems in brain tissue is now better understood with the concept of a neurovascular unit[10] proposed by the Stroke Progress Review Group at the National Institute of Neurological Disorders and Stroke. Although chronic cerebral hypoperfusion also induces neurovascular unit dysfunction[11], protection of the neurovascular unit is an important therapeutic target because dysfunction can precipitate further hypoperfusion[12], and because vascular endothelial cell (EC) damage, one of the components of the neurovascular unit, precedes ischemic demyelination[12]. If these can be normalized before disease onset and progression, an increase in white matter lesions may be prevented. Transplantation of ECs may be a therapeutic option because of the increase expected in collateral blood circulation and neuroprotection by extrinsic factors secreted from ECs, so-called "angiocrine signals". Here, we also focused on the organ specificity of vascular ECs since it is unknown which tissues or organs are the source of ideal vascular ECs for transplantation.

¹Department of Signal Transduction, Research Institute for Microbial Diseases, Osaka University, Suita, Osaka, Japan. ²Department of Neurosurgery, Osaka University Graduate School of Medicine, Suita, Osaka, Japan. ³World Premier Institute Immunology Frontier Research Center, Osaka University, Osaka, Japan. ⁴Integrated Frontier Research for Medical Science Division, Institute for Open and Transdisciplinary Research Initiatives (OTRI), Osaka University, Osaka, Japan. ⁵Center for Infectious Disease Education and Research, Osaka University, Osaka, Japan. ⁶These authors contributed equally: Yuichi Matsui, Fumitaka Muramatsu. ✉e-mail: ntakaku@biken.osaka-u.ac.jp

We have identified ECs having stem cell features in pre-existing blood vessels in each organ examined, especially in the liver. Such vascular endothelial stem-like cells are positive for the cell surface marker, CD157, and contribute to liver sinusoids in the long term. Endothelial cells derived from CD157-positive EC stem cells continuously secrete coagulation factor VIII and completely improve the bleeding phenotype observed in a hemophilia model in mice upon transplantation[13]. We showed that these cells are undifferentiated, monogenic, and represent several percent of all vascular ECs. Although we have reported that vascular endothelial stem cells reconstruct functional blood vessels in the drug-induced damaged liver and ischemic lower extremities[13,14], the effects of transplantation of such cells in chronic cerebral hypoperfusion are unknown. We tested the effect of transplanting vascular ECs into a mouse model of chronic cerebral hypoperfusion to prevent white matter lesions. We also evaluated the organ specificity of vascular ECs and the effect of vascular endothelial stem cells on chronic cerebral hypoperfusion as a proof of concept for future clinical application.

## Results

### Organ specificity of donor ECs

Vascular ECs were transplanted and evaluated in a mouse model of chronic cerebral hypoperfusion as described in Fig. 1a. In this model, blood flow in brain was attenuated up to 70% compared to in normal mice[15]. We isolated vascular ECs, designated by CD45 negative and CD31 positive (CD45−CD31+) staining, from several organs (tissues). When cultured in vitro, the proliferative capacity of primary ECs depends on their organ of origin. Endothelial cells derived from the heart, skin, or lung did not show higher proliferation rates compared to those from the liver or fat (Supplementary Fig. 1). Therefore, we determined that ECs from the brain, liver, and fat (brown) are appropriate for use in order to verify differences between donor organs in this cerebral hypoperfusion model.

We evaluated the formation of a vascular system by exogenously inoculated ECs using live imaging in two-photon microscopy. Our time-course data showed vascular recovery in the same tissue region in the same mice. Endothelial cells from the aforementioned organs formed capillary-like structures in the brain on day 7; however, apparently, brain-derived vascular ECs still formed a capillary-like network on day 21 compared to ECs from the other organs (Fig. 1b, c). The proliferation rate (from days 7 to 21) was approximately six times greater for ECs from the brain than ECs from the liver or fat (Fig. 1c).

In this model of chronic cerebral hypoperfusion, glial fibrillary acidic protein (GFAP)-positive activated astrocytes assembled and white matter lesions developed (Fig. 1d). As shown in Fig. 1d, e, white matter lesions were significantly less induced when brain-derived ECs were transplanted compared with ECs that originated from other organs, suggesting the lowest degree of cerebral ischemia was affected by the transplantation of brain-derived ECs. Staining for myelin basic protein was also performed to assess white matter lesions according to their degree of demyelination. Similarly, the lowest degree of cerebral ischemia was observed when brain-derived ECs were transplanted (Supplementary Fig. 2). Next, mice have a habit of spending more time exploring novel compared to familiar objects and this habit is better maintained when memory is preserved. By making observations on the exploration of familiar and novel objects, and calculating a novelty score, we compared memory performance in mouse models transplanted with ECs from different organs of origin. Novelty scores were also correlated with the degree of white matter lesions (Fig. 1f). We also performed Y-maze tests to assess memory function. Mice that were transplanted with brain-derived ECs showed better function (Supplementary Fig. 3). Therefore, we concluded that brain-derived vascular ECs were superior in all of the following points: angiogenesis, a reduction in white matter lesions, and in higher brain function (memory). This suggests that brain-derived ECs are suitable for transplantation in a disease model with chronic cerebral hypoperfusion.

### BBB function of donor ECs

Next, we analyzed whether ECs in newly developed blood vessels, after transplantation into the brains of model mice, showed a site-specific function, i.e., a BBB. This was examined by detecting the expression of BBB components such as claudin-5 (tight junction molecule), neural/glial antigen 2 (NG2; pericyte coverage), and glial fibrillary acidic protein (GFAP; astrocyte endofoot). It was found that GFP+ vascular tubs showed claudin-5 staining that, of the three sources, was highest in brain-derived ECs (Fig. 2a). Pericyte coverage was prominently observed in newly developed blood vessels generated after brain EC transplantation (Fig. 2b). This was confirmed by positive CD13 staining, another marker of pericytes (Supplementary Fig. 4). Astrocyte end-foot coverage was also significantly observed in ECs derived from the brain rather than from other sources (Fig. 2c), although GFAP-positive cells seemed to be abundant among ECs derived from liver ECs and around newly developed blood vessels.

Vessel imaging with fluorescent nanoparticles revealed good trafficking to donor vessels (newly developed GFP-positive ones) from recipient vessels in the case of brain-derived ECs (Fig. 2d). However, such trafficking was poor for liver-derived ECs and moderate for adipose-derived ECs (Fig. 2d). In addition, we injected tetramethylrhodamine-labeled dextran intravenously into each transplanted mouse and detected vascular permeability. Leakage was not observed from vessels generated by the transplantation of brain ECs. However, leakage occurred from vessels generated by the transplantation of fat or liver-derived ECs (Supplementary Fig. 5). Taken together, we concluded that brain-derived ECs generated adequate brain-specific blood vessels upon transplantation and that ECs from organs other than the brain may not fully function as cerebral vessels when transplanted.

### Comparison of neurotrophic and neuroprotective factor expression in ECs from different sources

It has been widely accepted that not only blood flow but also secreted molecules, so-called "angiocrine factors", from ECs affect the degree of cerebral ischemia[16]. Therefore, we next examined neurotrophic and neuroprotective factors in ECs from three different organs with the potential to influence the degree of cerebral ischemia. Of the molecules tested, brain-derived neurotrophic factor (BDNF), fibroblast growth factor (FGF)2, and platelet-derived growth factor (PDGF)-B were more abundant in brain-derived vascular endothelium, while nerve growth factor (NGF) and insulin-like growth factor (IGF)−1 were less abundant in the same cells (Fig. 3a). This indicates that the expression of humoral factors was also organ-specific and that BDNF is one of the candidates that protects against neuronal damage in our model.

Vascular endothelial growth factor (VEGF)-A affects both the proliferation of ECs and neuroprotection[17,18]. In our model, in addition to neuroprotection, the proliferative potential after transplantation was stronger in brain-derived ECs and we therefore evaluated VEGF-A in these cells. However, the expression of VEGF-A as well as VEGF receptor (VEGFR)−2 in ECs was not significantly different between ECs from various organs (Fig. 3b).

### Comparison of EC proliferation in vitro

In an in vivo setting, we were able to demonstrate the superiority of brain-derived ECs when examining the development of ECs in the brain; however, it remained undetermined whether proliferative capacity differed in ECs of different origin or whether the proliferation of ECs from liver or fat was restricted in a brain microenvironment. To assess these, we initially cultured ECs from different organs on OP9 stromal feeder cells, which are usually used for the organ culture of ECs. As shown in Fig. 4a in coculture with OP9 cells, the proliferative capacity of ECs from liver was highest among the three different sources. Next, in order to reproduce the environment of the brain in vivo, we planned to coculture with a cell population containing neurons and glial cells. In response to previous reports[19], we extracted cells from a fetal brain and attempted to coculture such cells with ECs derived from the three different organs. Interestingly, the proliferative potency of brain-

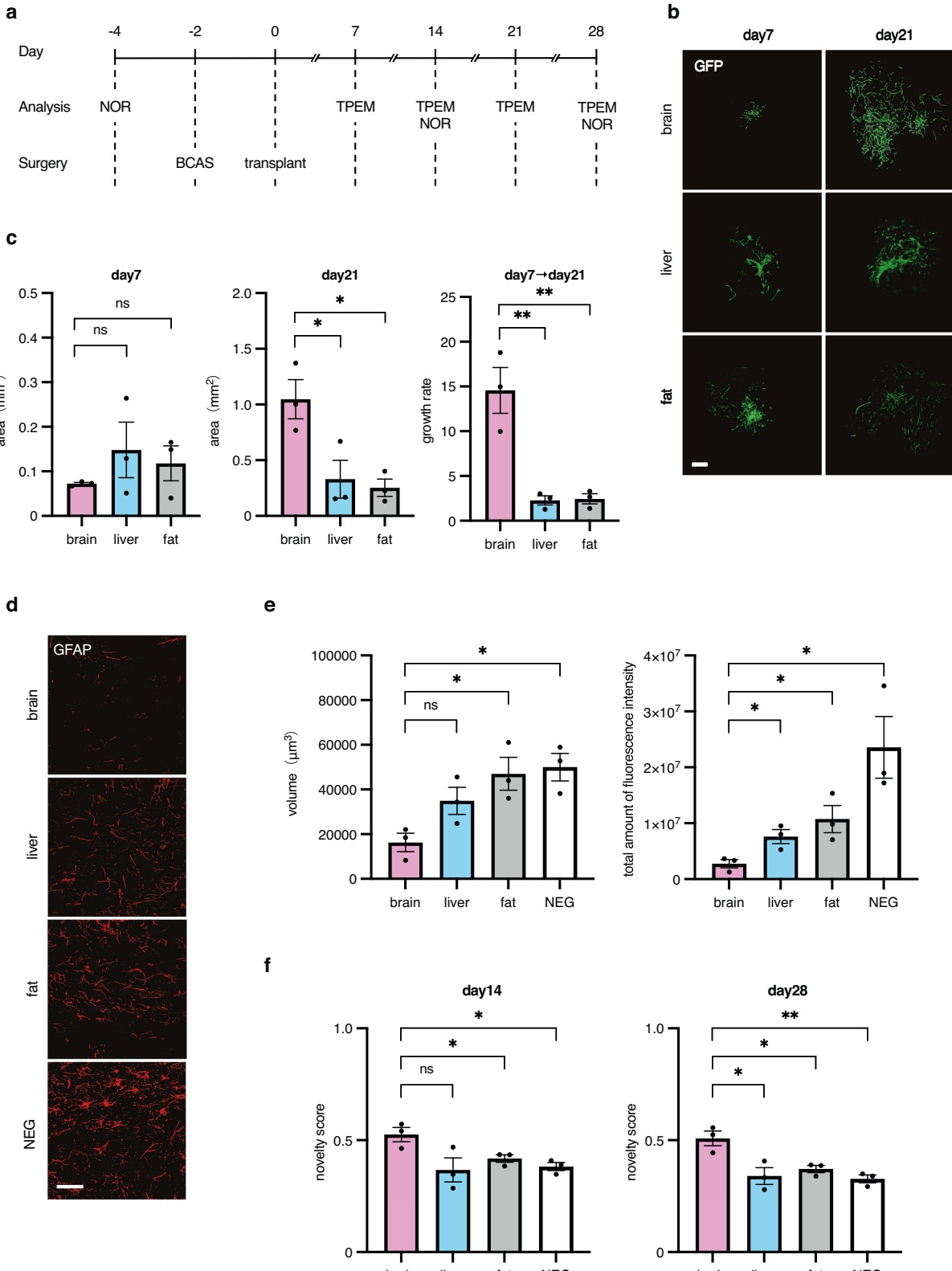

**Fig. 1 | Organ specificity of donor ECs. a** Outline of the transplantation experiment (**a–f**). **b** Representative images of the progress of transplanted ECs derived from C57BL/6-Tg (CAG-EGFP) mice in the brain. The endogenous fluorescence intensity of GFP was observed under TPEM. Scale bar, 500 μm. **c** Bar graph showing the mean of the regenerated vascular area by GFP-positive ECs (on days 7 and 21) and the growth rate (from days 7 to 21) of GFP-positive ECs (*n* = 3 per group). The growth rate was defined as the percentage increase in vascular area from days 7 to 21. Data are presented as mean ± SEM. **d** Representative images stained with anti-GFAP pAb (Alexa Flour 546) on day 28 after transplantation of ECs. Scale bar, 50 μm. **e** Bar graph showing the mean volume and total amount of fluorescence intensity of GFAP-positive cells (*n* = 3 per group) suggesting the degree of fibrosis. Data are presented as mean ± SEM. **f** Bar graph showing the mean novelty score on days 14 and 28 (*n* = 3 per group). Data are presented as mean ± SEM. *, *P* < 0.05; **, *P* < 0.01; ns not significant. BCAS bilateral common carotid artery stenosis, CAG-EGFP CAG-enhanced green fluorescent protein, EC endothelial cell, GFAP glial fibrillary acidic protein, GFP green fluorescent protein, NEG negative control mice received no cell transplants, NOR novel object recognition, ns not significant, pAb polyclonal antibody, TPEM two-photon excitation microscopy.

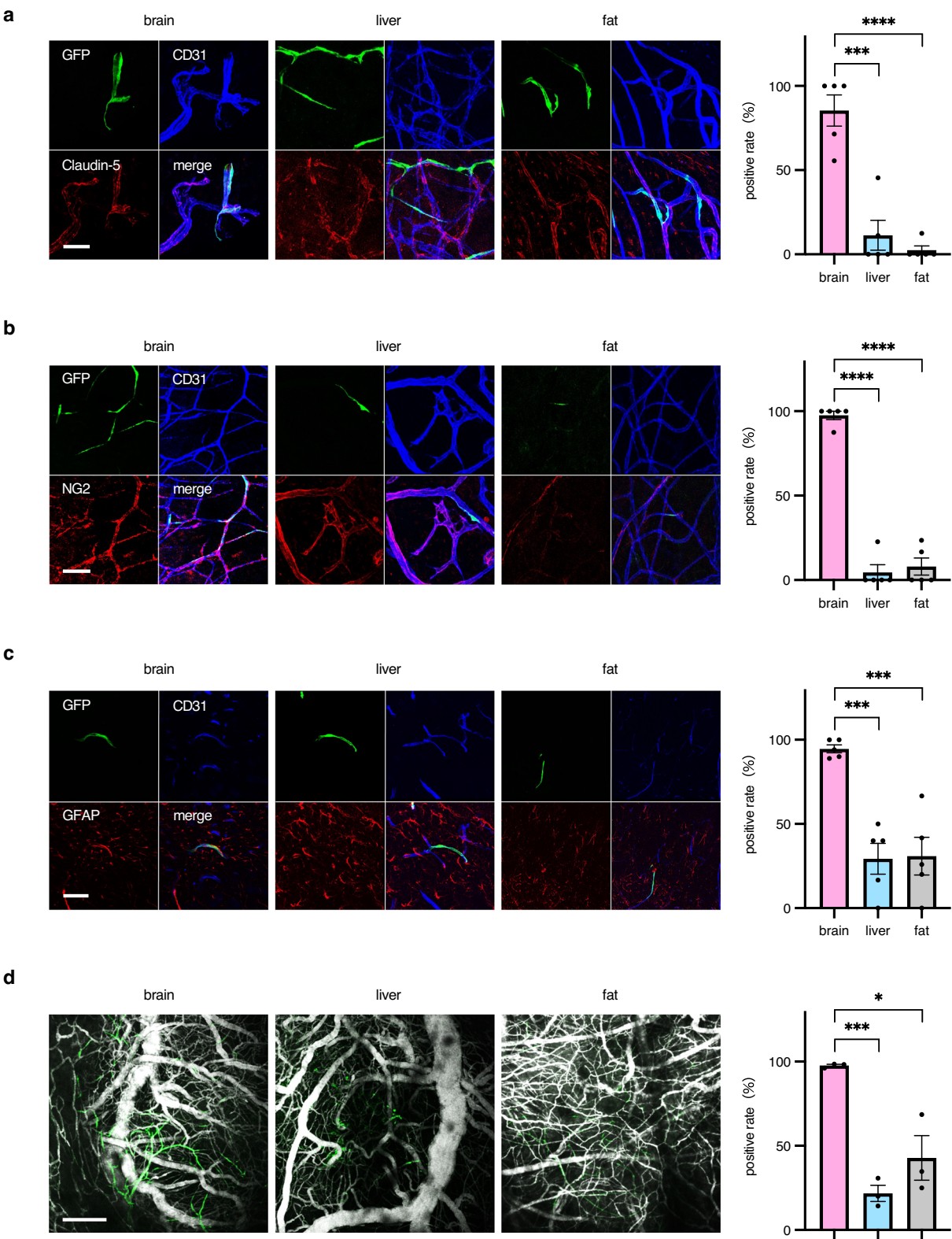

derived ECs increased; however, that of liver- and fat-derived ECs significantly decreased (Fig. 4b). This suggests that environmental molecular cues from the recipient, in this case the brain, induce brain-derived ECs to proliferate; however, these also suppress the proliferative ability of ECs from organs other than the brain. When culture supernatant from fetal brain cells was added, similar effects to those observed with the coculture of ECs with fetal brain cells were observed (Fig. 4c). It may be that direct cell–cell contact

is involved in the alteration of EC proliferation; also, it was apparent that soluble molecules from brain cells affect the proliferation of ECs. When astrocytes were isolated and cocultured with ECs from different organs, significant changes were not noted (Fig. 4d). Therefore, these observations suggest that secreted molecules from neuronal cells affected the proliferation of ECs. Moreover, in order to examine the effects of tissue-specific factors on the proliferation of organ-specific ECs, we performed coculture experiments

**Fig. 2 | Blood–brain barrier formation induced by brain-derived ECs.** Vascular phenotypes in brains transplanted with ECs from different organs were analyzed on day 21 as described in Fig. 1a. **a** Representative images show staining with anti-CD31 mAb (Alexa Fluor 647) and anti–claudin-5 pAb (Alexa Fluor 546) in regenerated vasculature transplanted with GFP-positive ECs derived from organs as indicated. The positive rate refers to the proportion of claudin-5–positive ECs among GFP-positive ECs expressed as a percentage. Scale bar, 30 μm. *n* = 5 per group. Data are presented as mean ± SEM. **b, c** Representative images stained with anti-CD31 mAb (Alexa Fluor 647), anti-NG2 pAb (Alexa Flour 546) (**b**), anti-CD31 mAb (Alexa Flour 647), and anti-GFAP pAb (Alexa Fluor 546) (**c**) in regenerated vasculature areas transplanted with GFP-positive ECs derived from organs as indicated. The positive rate refers to the proportion of NG2-positive pericytes covering ECs (**b**) or astrocyte end-feet (GFAP-positive) positive ECs (**c**) among GFP-positive ECs expressed as a percentage. Scale bar, 30 μm. *n* = 5 per group. Data are presented as mean ± SEM. **d** Fluorescence images of vasculature transplanted with GFP-positive ECs derived from various organs as indicated. Blood flow was overlapped by labeling with AngioSPARK 680. The positive rate refers to the proportion of GFP-positive vessels with blood flow among GFP-positive ECs expressed as a percentage. Scale bar, 300 μm. *n* = 3 per group. Data are presented as mean ± SEM. *, $P < 0.05$; ***, $P < 0.001$; ****, $P < 0.0001$. EC endothelial cell, GFAP glial fibrillary acidic protein, GFP green fluorescent protein, mAb monoclonal antibody, NG2 neural/glial antigen-2, pAb polyclonal antibodies.

using cells derived from the liver or fat with primary ECs from the brain, liver, or fat. We sorted CD31⁻ CD45⁻ cells derived from fat or the liver and cocultured these with ECs on OP9 feeder cells. The proliferation of vascular ECs differed when cocultured with cells from different organs as partner cells (Supplementary Fig. 6). In cocultures with brain-derived cells, ECs derived from the same organ, the brain, proliferated but the growth of ECs from different organs decreased. However, cocultures with liver- or fat-derived ECs did not show similar findings to those of brain-derived ECs. Specifically, cocultures with liver-derived cells resulted in the decreased proliferation of liver-derived ECs and the increased proliferation of brain- and fat-derived ECs. In cocultures with fat-derived cells, the proliferation of fat-derived ECs increased but brain-derived ECs also increased while liver-derived ECs decreased.

### Netrin-1 from brain cells alters the proliferation of ECs

Based on findings of the coculture of ECs with brain cells above, we tried to isolate molecules affecting the proliferation of ECs. Since angiogenesis and axon formation are closely related and axon guidance molecules affect angiogenesis[20], we predicted netrin was a candidate for EC proliferation. The netrin family consists of secreted netrins 1–4 and membrane-bound netrin, G1-2[21]. When netrin-1, a secreted netrin, was added to a coculture of ECs from different organs on OP9 feeder cells, growth was suppressed in liver or fat ECs. By contrast, the proliferation of ECs from the brain was promoted by netrin-1 in a dose-dependent manner (Fig. 5a). These results were similar to those obtained with the addition of fetal brain cells. To verify whether EC proliferation would occur similarly in vivo, a Matrigel plug assay was performed whereby ECs derived from each organ were mixed in Matrigel with netrin-1 and injected subcutaneously into mice. In Matrigel plugs with brain-derived ECs, netrin-1 induced EC proliferation; such an effect was not observed in gels with liver- or fat-derived ECs (Supplementary Fig. 7). Next, we inhibited netrin-1 with anti–netrin-1 antibody in a coculture of ECs with fetal brain cells. Correlating with the concept of netrin-1 affecting cell proliferation, anti–netrin-1 antibody abrogated brain cell–induced proliferation of brain-derived ECs in vitro. By contrast, in liver- and fat-derived ECs, fetal brain cells induced the suppression of EC growth that was canceled by anti–netrin-1 antibody (Fig. 5b). Netrin-1 affects cell kinetics depending on the expression of different receptors. Therefore, we next evaluated the expression of netrin-1 receptors in ECs from brain, fat, and liver. We found that brain-derived ECs specifically expressed Unc 5B compared to ECs from other organs and that this difference may affect the proliferation of ECs (Fig. 5c). To examine whether the netrin-1 and Unc 5B systems confer brain-derived ECs with organ specificity, ECs derived from the liver were transduced with Unc 5B–expressing lentiviral vectors and we then examined the effects of netrin-1 on their proliferation. Similar to that of brain ECs, the proliferation of Unc 5B–expressing liver ECs was enhanced by netrin-1 (Supplementary Fig. 8). This suggests that the netrin-1/Unc 5B system generally functions when ECs express Unc 5B.

Based on these results, we next attempted to neutralize fetal brain cell–derived netrin-1 by adding the soluble receptor, Unc 5B. We found that the effect of fetal brain cells on brain-derived ECs was alleviated, but no change was observed in liver- and fat-derived ECs (Fig. 5d). It was thus suggested that netrin-1 and Unc 5B may be involved in the proliferative specification of brain-derived ECs.

### Transplantation of CD157-positive ECs

We previously reported that CD157-positive ECs are stem cell–like cells[13] and hypothesized that transplantation of these cells would be more effective against chronic cerebral hypoperfusion. We tested this hypothesis by comparing brain-derived CD157-positive and -negative ECs. In vitro, CD157-positive ECs had higher proliferative potential than CD157-negative ECs as observed in ECs from the liver[13] (Fig. 6a). CD157-positive ECs derived from each organ were cultured in the presence of netrin-1. Similarly, as observed in Fig. 5, the proliferation of liver or fat ECs was inhibited but that of brain ECs was enhanced by netrin-1 (Supplementary Fig. 9). Next, a comparison was made by the transplantation of ECs in a chronic cerebral hypoperfusion model. CD157-negative ECs barely contributed to neovascular system formation; however, CD157-positive ECs showed high proliferative potency even in the chronic hypoperfused brain (Fig. 6b, c). In addition to higher proliferative capacity, CD157-positive ECs were superior in reducing white matter formation compared to CD157-negative ECs (Fig. 6d, e) and in mice retaining memory (Fig. 6f). Regarding the expression of brain-protective humoral factors, such as BDNF, FGF2, and PDGF-B, we found that BDNF expression was higher in CD157-positive compared to CD157-negative ECs (Supplementary Fig. 10). These results suggest that the use of brain-derived and CD157-positive ECs for EC transplantation may provide a more effective therapeutic effect in chronic cerebral hypoperfusion.

### Discussion

The present study suggested that the transplantation of vascular ECs reduced white matter lesions and also reduced brain dysfunction, i.e., memory impairment, in chronic cerebral hypoperfusion. However, these effects were observed when brain-derived vascular ECs, but not fat- or liver-derived ECs, were used. Vascular ECs are known to differ in morphology and function depending on the organ[22] but differences in the dynamics of vascular ECs from each organ within the environment of the brain are unknown. When transplanted into the liver, liver-derived ECs were shown to undergo tissue-specific differentiation (sinusoids) in our previous study[13]. It is well known that no significant difference exists in VEGF expression between brain and liver ECs[23]. It was also reported that VEGFR-2 expression and the expression of VEGF signaling–related genes are higher in liver ECs compared to brain ECs. Regarding the proliferation of engrafted ECs, it is thought that the effect of cell–cell interactions at the transplant site is greater than the effect of differences in the expression level of VEGF signaling–related genes in transplanted ECs. Neuronal axons and blood vessels often follow the same pathways when inducing each other[24]. Vessels induce axons by generating artemin, neurotrophin 3, and other molecules[25,26] and, conversely, nerves induce vessels by generating VEGF and other molecules[27]. Axon guidance molecules induce not only axon elongation but also angiogenesis[24]; as examples, netrin, slit, semaphorin, and ephrin have been identified in the past[28].

In this study, we focused on netrin and identified netrin-1, a secreted form of netrin, as a candidate factor involved in the mechanism by which brain-derived vascular ECs promoted angiogenesis in the brain as

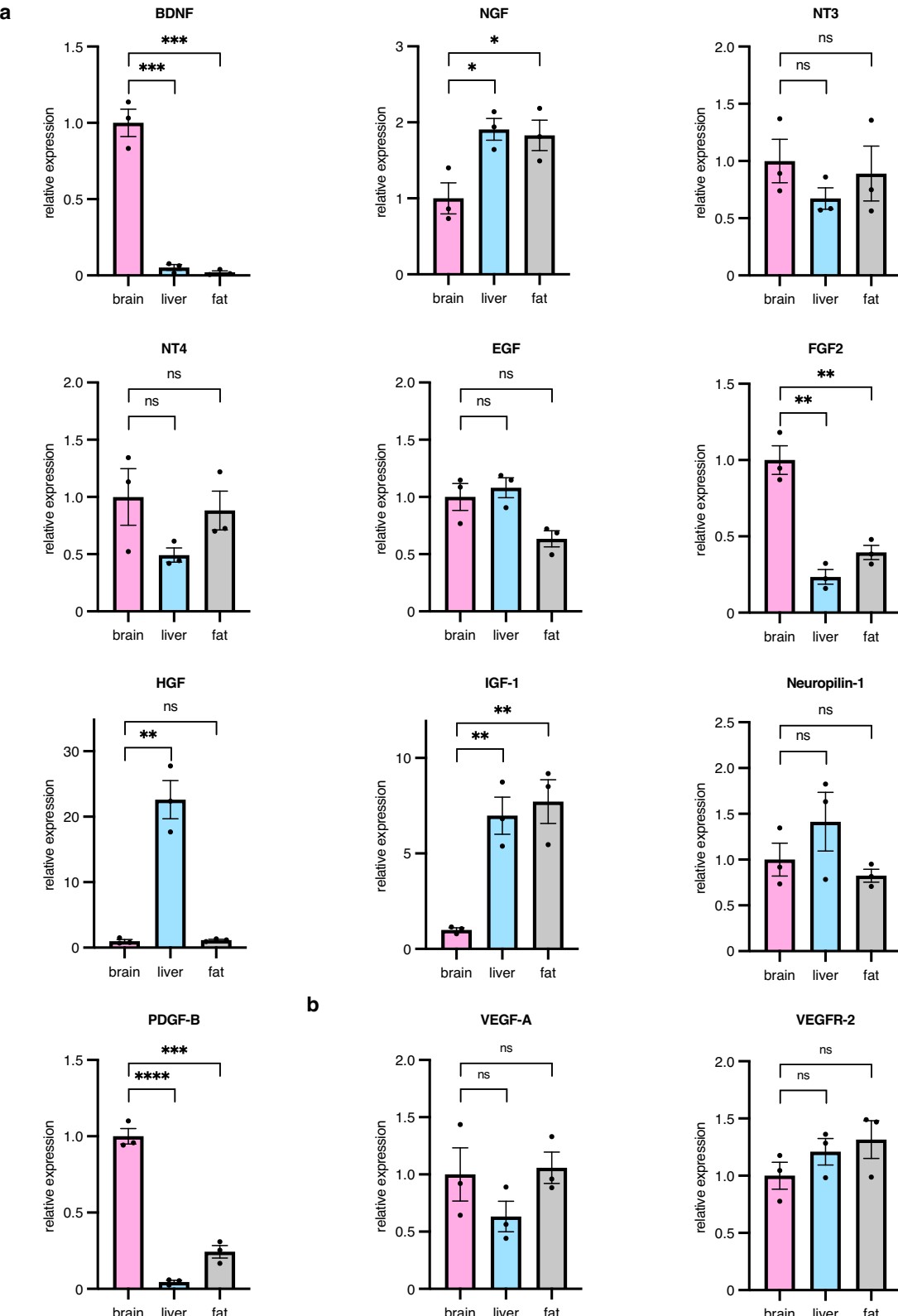

**Fig. 3 | Humoral factors from ECs of different origins. a** Quantitative reverse transcription PCR analysis of neurotrophic and neuroprotective factors (BDNF, NGF, NT3, NT4, EGF, FGF2, HGF, IGF-1, neuropilin-1, and PDGF-B) in ECs from brain, liver, and fat (*n* = 3 per group). The gene expression of factors is relative to that of brain-derived ECs, which is set to 1. Data are presented as mean ± SEM. **b** Quantitative reverse transcription PCR analysis for angiogenesis-related factors (VEGF-A and VEGFR-2) in ECs from brain, liver, and fat (*n* = 3 per group). The gene expression of factors is relative to that of brain-derived ECs, which is set to 1. Data are presented as mean ± SEM. *, *P* < 0.05; **, *P* < 0.01; ***, *P* < 0.001; ****, *P* < 0.0001; ns not significant. BDFN brain-derived neurotrophic factor, EC endothelial cell, EGF epidermal growth factor, FGF2 fibroblast growth factor 2, HGF hepatocyte growth factor, IGF-1 insulin-like growth factor-1, NGF nerve growth factor, NT neurotrophin, PDGF-B platelet-derived growth factor-B, VEGF vascular endothelial growth factor, VEGFR vascular endothelial growth factor receptor.

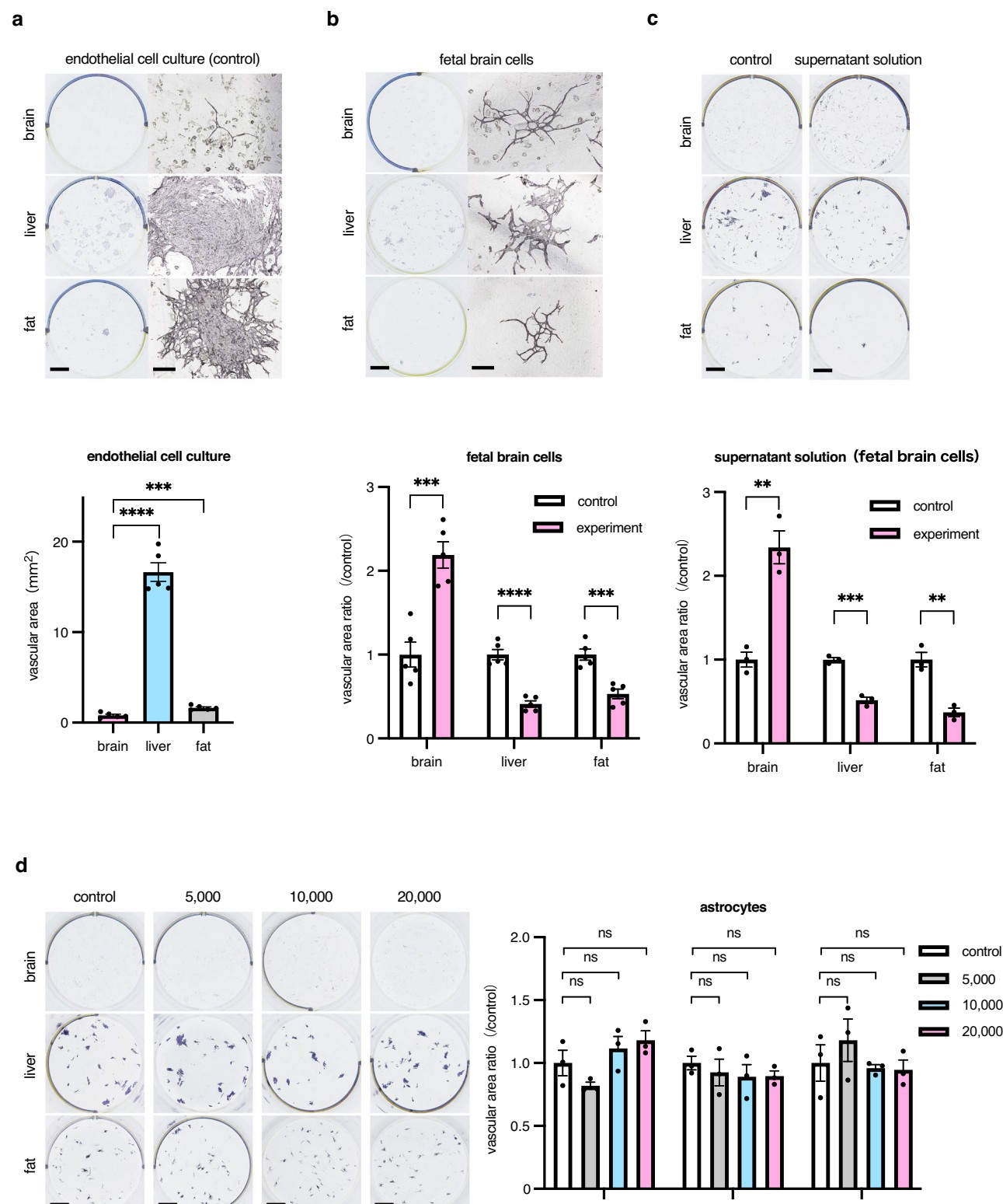

previously suggested[29,30]. Moreover, treatment of mouse models of acute cerebral ischemia with a combination of netrin-1 and isoflurane induced angiogenesis and neurological recovery by activating the hypoxia-inducible factor-1α−netrin-1−UNC5B/VEGF cascade[31]. Angiogenesis in glioblastoma is also promoted by netrin-1[32,33]. A dual role for netrin-1 in angiogenesis has also been reported, with netrin-1 at low concentrations (500 ng/mL or less) acting in a promoting manner in the presence of Unc 5B, and at high concentrations (1000 ng/mL or more) in an inhibitory manner[34]. Another group reported that Unc 5B acts in an inhibitory manner in the induction of endothelial tip cells[35]. CD146 is an angiogenesis-promoting receptor for netrin-1[36] and the angiogenesis-promoting activity of netrin-1 is not mediated by any known netrin-1 receptor[37], indicating that the role of netrin in angiogenesis is very complex. Further studies are needed to elucidate the detailed mechanism of netrin-1−induced angiogenesis. However, in this

**Fig. 4 | EC proliferation in vitro. a** Representative photographs of EC colonies. $5 \times 10^3$ ECs isolated from each organ as indicated were cultured on OP9 feeder cells for 10 days and stained with anti-CD31 antibody. Scale bar, 3 mm (low-power field) and 200 μm (high-power field). The mean vascular area generated by ECs from each organ ($n = 5$ per group) was quantified. Data are presented as mean ± SEM. **b** Representative photographs of EC colonies. $5 \times 10^3$ ECs isolated from each organ as indicated were cultured with or without (control) fetal brain cells on OP9 feeder cells for 10 days and stained with anti-CD31 antibody. Scale bar, 3 mm (low-power field) and 200 μm (high-power field). The mean vascular area generated by ECs from each organ ($n = 5$ per group) was evaluated. The vascular area with brain cells was compared to that without (control) brain cells. Data are presented as mean ± SEM. **c** Representative photographs of EC colonies. $5 \times 10^3$ ECs isolated from each organ as

indicated were cultured with or without (control) the culture supernatant of fetal brain cells on OP9 feeder cells for 10 days and stained with anti-CD31 antibody (**d**). Scale bar, 3 mm. The mean vascular area generated by ECs from each organ ($n = 5$ per group) was evaluated. The vascular area with the culture supernatant of brain cells were compared to that without supernatant. Data are presented as mean ± SEM. **d** Representative photographs of EC colonies. $5 \times 10^3$ ECs isolated from each organ as indicated were cultured with or without (control) astrocytes from fetal brain cells on OP9 feeder cells for 10 days and stained with anti-CD31 antibody. Scale bar, 3 mm. The mean vascular area generated by ECs from each organ ($n = 5$ per group) was evaluated. The vascular area generated with astrocytes were compared to that without astrocytes. Data are presented as mean ± SEM. **, $P < 0.01$; ***, $P < 0.001$; ****, $P < 0.0001$; ns not significant. EC endothelial cell.

study, we found that brain-derived vascular ECs expressed more Unc 5B than those from other organs. Moreover, angiogenesis was activated by netrin-1, suggesting that netrin-1 and Unc 5B systems may be involved in the formation of a brain-specific vascular system.

Differences in the expression of neurotrophic and neuroprotective humoral factors among organs may also influence ischemia reduction. In the present study, BDNF, FGF2, and PDGF-B were highly expressed in brain-derived vascular ECs. This result is consistent with a report[16] that vascular ECs secrete organ-specific cytokines, known as angiocrine factors, that aid in organ recovery during injury. Similarly, with this report, it was revealed that cognitive dysfunction was reduced via BDNF[38–40]. Since BDNF expression is reduced in the brain during chronic cerebral hypoperfusion[39], supplementation with BDNF-rich ECs makes sense from the perspective of replenishing what is missing. It has been shown that FGF2 is up-regulated in the brain and induces radial glial cells under conditions of chronic cerebral hypoperfusion[41]; radial glial cells play an important role in neurogenesis. An FGF2-enriched environment enhances endogenous angiogenesis and the function of motor neurons in chronic cerebral hypoperfusion brain injury[42]. Moreover, the administration of protein nanoparticles modified with PDGF-B reduced the extent of acute cerebral infarction and improved neurological function[43]. Further studies are needed to elucidate detailed mechanisms but it is reasonable to conclude that angiocrine signals, i.e., BDNF, FGF2, and PDGF-B, from brain ECs are involved in improving chronic cerebral hypoperfusion.

Because diversity and heterogeneity exist in adult vascular ECs, it has been proposed that tissue-endemic vascular endothelial stem cells may exist[44–48]. We have identified and reported on vascular ECs expressing CD157 that exhibit special characteristics[13]. CD157-positive ECs have high proliferative potential and the ability to differentiate into CD157-negative cells. They are capable of reconstructing blood vessels and can produce multiple CD157-positive cells after transplantation of a single cell, making them unique as stem cells for vascular ECs. Transplantation of CD157-positive ECs into ischemic sites can also be expected to form new blood vessels and improve blood flow. From the angiocrine factor perspective described above[16], transplantation of endothelial stem cells may contribute not only to the formation of new blood vessels but also to the recovery of organ function. In fact, orthotopic transplantation of endothelial stem cells from liver or limb muscle into liver injury or lower limb ischemia models, respectively, has been shown to promote organ recovery[13,14]. In the present study, CD157-positive ECs from brain transplanted into a chronic cerebral hypoperfusion mouse model showed a higher capacity for angiogenesis, fewer white matter lesions, and less higher brain dysfunction compared to CD157-negative ECs. As a humoral factor, high expression of BDNF was also observed. We already reported that CD157-positive ECs expanded after transplantation into liver, suggesting that BDNF from CD157-positive ECs may also contribute to tissue recovery.

The transplantation of endothelial stem cells for cerebral ischemia is a promising treatment in clinical practice, but the problem is that only about 5% of all vascular ECs in the brain are CD157-positive[13]. Therefore, the absolute number of ECs is too small and the use of these cells from a donor is not practical. Although the autologous transplantation of adipose-derived endothelial stem cells would be clinically applicable, the results of our

present study indicate that brain-derived vascular ECs can induce excellent tissue recovery in ischemic brains. It is possible that the transplantation of endothelial stem cells of an undifferentiated status results in superior tissue recovery. Therefore, the in vitro or ex vivo expansion of endothelial stem cells is desirable to realize endothelial stem cell transplantation as a viable treatment for cerebral ischemia in future.

## Methods
### Mice
C57BL/6 mice and C57BL/6-Tg (cytomegalovirus enhancer fused to the chicken beta-actin promoter [CAG]–enhanced green fluorescent protein [EGFP]) mice (which express EGFP in all tissues) were purchased from Japan SLC (Shizuoka, Japan). All animals were housed in a temperature-controlled environment with a 12-h light/dark cycle, and food and water were freely provided. All experimental procedures in this study were approved by the institutional Animal Care and Use Committee of Osaka University.

### Chronic cerebral hypoperfusion mouse model
Chronic cerebral hypoperfusion was induced by bilateral common carotid artery (CCA) stenosis using a microcoil in male C57BL/6 mice (8–10 weeks old) as described previously[15]. Briefly, mice were anesthetized with three mixed anesthetic agents consisting of 0.3 mg/kg of medetomidine (Domitor, Nippon Zenyaku Kogyo, Fukushima, Japan), 4.0 mg/kg of midazolam (Sandoz, Holzkirchen, Germany), and 5.0 mg/kg of butorphanol (Vetorphale, Meiji Seika Pharma, Tokyo, Japan). Bilateral CCAs were exposed through a midline cervical incision. The right CCA was gently lifted and placed between the loops of a microcoil (inner diameter 0.18 mm, pitch 0.5 mm, total length 2.5 mm; purchased from Samini, Hamamatsu, Japan) below the carotid bifurcation. The microcoil was twined by rotating it around the CCA. After 30 min, the same procedure was conducted to the left CCA. The anesthetic was reversed with 0.3 mg/kg of atipamexole (Antisedan, Nippon Zenyaku Kogyo).

### Cell preparation and flow cytometry
Cells from C57BL/6 and C57BL/6-Tg (CAG-EGFP) mice were isolated and cell-surface antigen staining was performed as previously described in ref. [13]. Briefly, mice were euthanized, and organs were excised, minced, and digested with dispase II (Thermo Fisher Scientific), collagenase (Wako, Osaka, Japan), and type II collagenase (Worthington Biochemical Corp., Lakewood, NJ, USA) with continuous shaking at 37 °C. The digested tissue was filtered (40-μm filters) to obtain single-cell suspensions. Cell surface antigen staining was performed with anti-CD31 (clone MEC13.3, BD Biosciences, San Diego, CA, USA), anti-CD45 (clone 30-F11, BD Biosciences), and anti-CD157 (clone BP3, Biolegend) antibodies. The stained cells were analyzed and sorted by a SOAP FACSAria (BD Bioscience) and data were analyzed by FlowJo Software (Treestar Software, San Carlos, CA, USA).

### EC transplantation
Endothelial cells ($3\times10^5$ cells/well) from C57BL/6-Tg (CAG-EGFP) mice (5–6 weeks old) were shaken overnight in HuMedia-EB2 supplemented with HuMedia-EG (Kurabo, Osaka, Japan) and 2% B-27 supplement

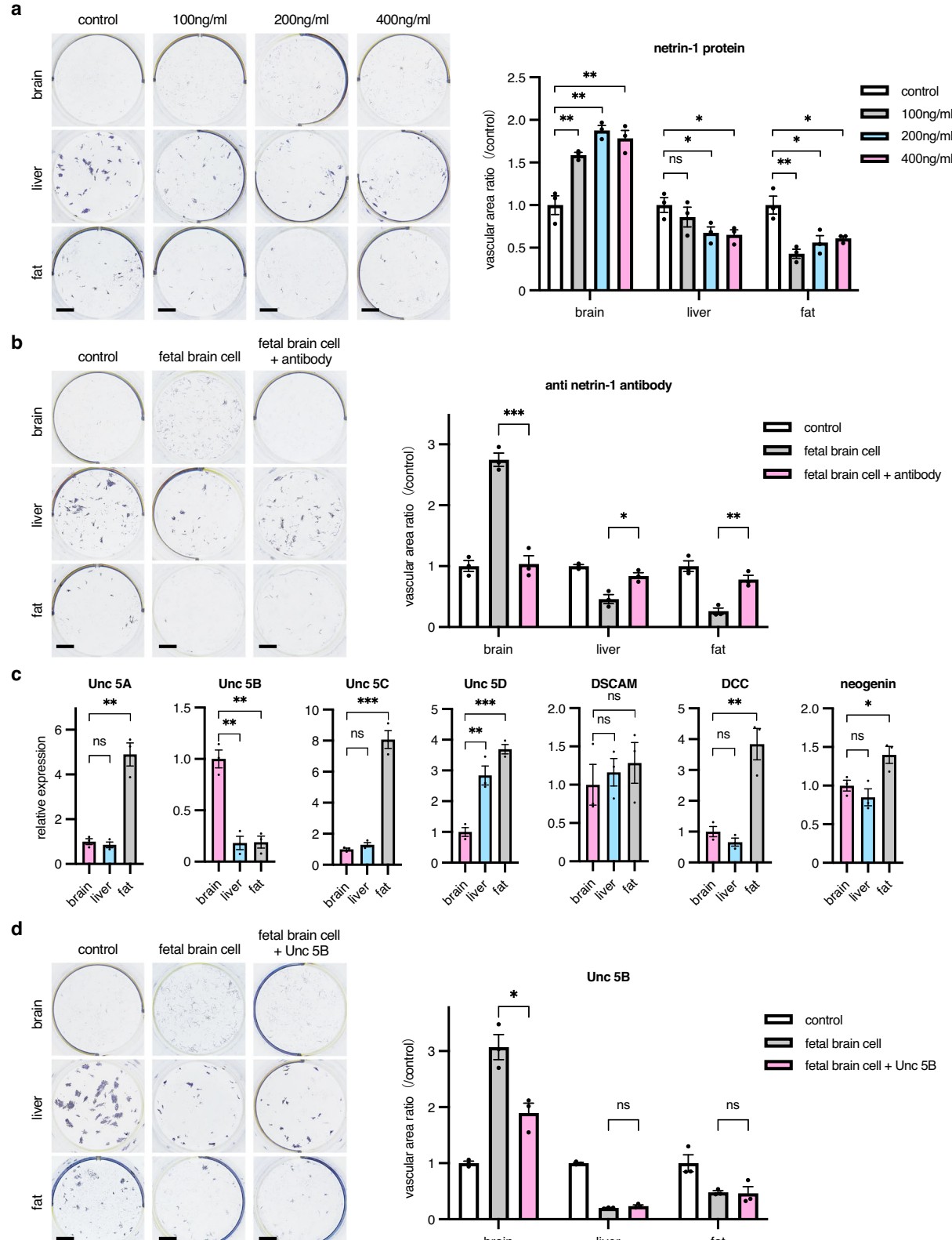

(Thermo Fisher Scientific, Waltham, MA, USA), and transplanted into a chronic cerebral hypoperfusion brain model. Specifically, model mice were anesthetized with three mixed anesthetic agents consisting of 0.3 mg/kg of medetomidine, 4.0 mg/kg of midazolam, and 5.0 mg/kg of butorphanol. To prevent cerebral edema during surgery, mice were injected intraperitoneally with 300 μL of 10% glycerin (Glyceol; Chugai Pharmaceutical, Tokyo, Japan). Holes were drilled in the skulls of model mice to expose the dura

matter, which was removed to expose the right cerebral cortex. Endothelial cells ($3 \times 10^5$) were stereotactically injected into a site, 3 mm lateral and 1 mm posterior from the bregma, to a depth of 60 μm. The transplanted areas were filled with artificial cerebrospinal fluid (Artcereb; Otsuka Pharmaceutical Factory, Tokushima, Japan) and sealed with a silicon-coated 5-mm round glass coverslip (Matsunami Glass Ind., Osaka, Japan) using a medical adhesive (Aron Alpha A; Daiichi Sankyo, Tokyo, Japan). Mice were

**Fig. 5 | Effect of netrin-1 on organ-specific EC proliferation. a** $5 \times 10^3$ ECs from each organ as indicated were cultured on OP9 feeder cells with several doses of netrin-1 protein and stained with anti-CD31 antibody. Control represents no netrin-1 addition. Representative images are shown on the left. Scale bar, 3 mm. Quantitative evaluation is shown on the right. In each group, values were compared to the no addition of netrin-1 ($n = 3$ per group). Data are presented as mean ± SEM. **b** $5 \times 10^3$ ECs from each organ as indicated were cultured on OP9 feeder and fetal brain cells, with and without anti–netrin-1 antibody, and stained with anti-CD31 antibody. Representative images are shown on the left. Scale bar, 3 mm. Quantitative evaluation is shown on the right. In each group, values were compared to no addition of brain cells or netrin-1 ($n = 3$ per group). Data are presented as mean ± SEM.

**c** Quantitative reverse transcription PCR analysis for netrin receptors (Unc 5A, Unc 5B, Unc 5C, Unc 5D, down syndrome cell adhesion molecule [DSCAM], deleted in colorectal cancer [DCC], and neogenin) in ECs from brain, liver, and fat ($n = 3$ per group). Expression was relative to that of brain ECs. Data are presented as mean ± SEM. **d** $5 \times 10^3$ ECs from each organ as indicated were cultured on OP9 feeder and fetal brain cells, with and without soluble Unc 5B, and stained with anti-CD31 antibody. Representative images are shown on the left. Scale bar, 3 mm. Quantitative evaluation is shown on the right. In each group, values were compared to the no addition of brain cells or soluble UNC5B ($n = 3$ per group). Data are presented as mean ± SEM. *, $P < 0.05$; **, $P < 0.01$; ***, $P < 0.001$; ns not significant. EC endothelial cell.

kept on a heated pad set to 37 °C until adhesion between the glass and skull bone was complete, and then the anesthetic was reversed with 0.3 mg/kg of atipamezole.

CD157-positive and -negative ECs ($2 \times 10^4$ cells/well) were also collected and transplanted in the same manner.

### In vivo visualization of transplanted vessels
In vivo visualization of transplanted vessels was performed with a TCS/SP8 confocal microscope (Leica Microsystems, Wetzlar, Germany) equipped with a Chamelon laser (Coherent, Santa Clara, CA, USA) as previously reported[49]. During imaging, an animal's head was immobilized using a custom-made cranial window holder. Mice were kept on a heated pad set to 37 °C, and anesthesia was maintained with isoflurane vaporizer and scavenger (Muromachi Kikai, Tokyo, Japan). In experiments to observe the area and growth rate of transplanted vessels, mice were observed 7, 14, 21, and 28 days after transplantation. In an experiment to observe the relationship with recipient vessels, mice were observed 21 days after transplantation and were injected retro-orbitally with 100 µL of AngioSPARK 680 (PerkinElmer, Waltham, MA, USA).

### Immunohistochemical staining
The procedure for tissue preparation and staining was as previously reported[50]. Briefly, fixed specimens were embedded in optimal cutting temperature compound (Sakura Finetek, Tokyo, Japan) and sectioned at 20 or 50 µm. For immunohistochemistry, the following antibodies were used: Armenian hamster anti-mouse CD31 mAb (MAB1398Z; Merck Millipore, Darmstadt, Germany; dilution 1/200), rabbit anti–claudin-5 pAb (polyclonal antibody; Abcam, Cambridge, MA, USA; dilution 1/200), rabbit anti–neural/glial antigen-2 (NG2) pAb (AB5320, Sigma–Aldrich, St. Louis, Missouri, USA; dilution 1/250), rabbit anti-glial fibrillary acidic protein (GFAP) pAb (HPA056030; Sigma–Aldrich; dilution 1/1000), Alexa Fluor 546–conjugated goat anti-rabbit IgG pAb (A-11010; Thermo Fisher Scientific; dilution 1/200), and Alexa Fluor 647–conjugated goat anti-Armenian hamster IgG pAb (127-605-160; Jackson ImmunoResearch Laboratories, West Grove, PA, USA; dilution 1/400). The sections were visualized using a Leica TSC SP5 confocal microscope and processed with Leica Application Suite (Leica Microsystems, Wetzlar, Germany) and Adobe Photoshop CC software (Adobe Systems, Mountain View, CA, USA).

### White matter lesions
Twenty-eight days after transplantation, each brain was removed, tissue 1.5 mm anterior and 2.5 mm posterior to the bregma region was cut out, and 20-µm slices were stained as described above. The volume and total amount of fluorescence intensity were measured as white matter lesions in GFAP-positive cells in the median of the vastus medullaris of the corpus callosum. Three average values per individual (separate sections) were measured.

### Functional assessment of BBB
Twenty-one days after transplantation, each brain was removed, the transplanted portion was cut out, and 50-µm slices were stained as described above. For claudin-5 and NG2 staining, GFP-positive cells expressing these antigens were defined as positive cells. For GFAP staining, GFP-positive

cells that were even partially covered by an endofoot were defined as positive cells. Positive cells were counted visually.

### Endothelial colony–forming assay
Primary ECs were isolated as above and $5 \times 10^3$ cells/well were cocultured with OP9 stromal cells ($2 \times 10^4$ cells/well; RIKEN cell bank, Tsukuba, Japan) in 24-well plates. Cultures were maintained in RPMI-1640 (Sigma–Aldrich) supplemented with 10% fetal calf serum (FCS; Sigma–Aldrich), 0.1% 2-mercaptoethanol (Gibco, Grand island, NY, USA) and 1% penicillin–streptomycin solution (Sigma–Aldrich). VEGF (10 ng/mL; PeproTech, Rocky Hill, NJ, USA) was added every 3 days. Fetal brain cells were harvested from C57BL/6 mice at embryonic day 14.5 and treated with accutase (Innovative Cell Technologies, San Diego, CA, USA) at 37 °C for 3 min. Fetal brain cells ($2 \times 10^4$ cells/well) were cultured in DMEM/F-12 (Gibco), 10% FCS, 2% B-27 supplement, and 1% penicillin–streptomycin solution for 1 week and then cocultured with ECs. Supernatant solution was taken from the medium in which the above fetal brain cells were cultured for 1 day and was added every 3 days. Astrocytes for coculturing with ECs were purchased from Lonza (Walkersville, MD, USA). Netrin-1 protein (R&D Systems, Minneapolis, MN, USA) was added every 3 days. Anti–netrin-1 antibody (10 µg/mL; R&D Systems) or Unc 5B (300 ng/mL; R&D Systems) was added every 3 days to cocultures of fetal brain cells. Endothelial cells were cultured for a total of 10 days. The procedure for staining was as previously reported[14]. For immunohistochemistry, an anti-CD31 antibody was used for staining and a biotin-conjugated polyclonal anti-rat IgG (Agilent Technologies, Santa Clara, CA, USA) was used as a secondary antibody. Biotinylated secondary antibodies were developed using ABC kits (Vector Laboratories, Newark, CA, USA). Samples were visualized using a Canon PowerShot SX50 HS camera (Canon, Tokyo, Japan) for low-powered fields, and a Leica DMi8 for high-powered fields.

### Image analysis
Images for vessels were analyzed using ImageJ software (National Institute of Health, Bethesda, MD, USA), and white matter lesions using Volocity software (Perkin Elmer, Waltham, MA, USA).

### Novel object recognition test
A novel object recognition (NOR) test, which follows a previously stated protocol[51] with minimal adjustment, was used to test the recognition memory of all animals. Briefly, mice were individually habituated to a 30 cm × 30 cm × 30 cm black plastic open-field box for 15 min on the day before the test. Mice were exposed to two novel light blue triangular wooden blocks A-A (5 cm × 3 cm × 3 cm height) symmetrically placed 10 cm from the nearest walls. The mice were placed equidistant from the two objects and allowed to explore for 15 min. Ninety minutes later, configuration A-A was changed to configuration A-B to assess memory by replacing one of the novel red wooden cylinders (3 cm height, 3 cm diameter). To minimize potential confounding spatial memory in contrast with object recognition, the animals were placed in the same corner as in their prior trial. Behavior was recorded for 15 min. The times spent exploring the familiar (F) object and the new (N) object were recorded for 15 min and assessed by an independent grader. A novelty score was

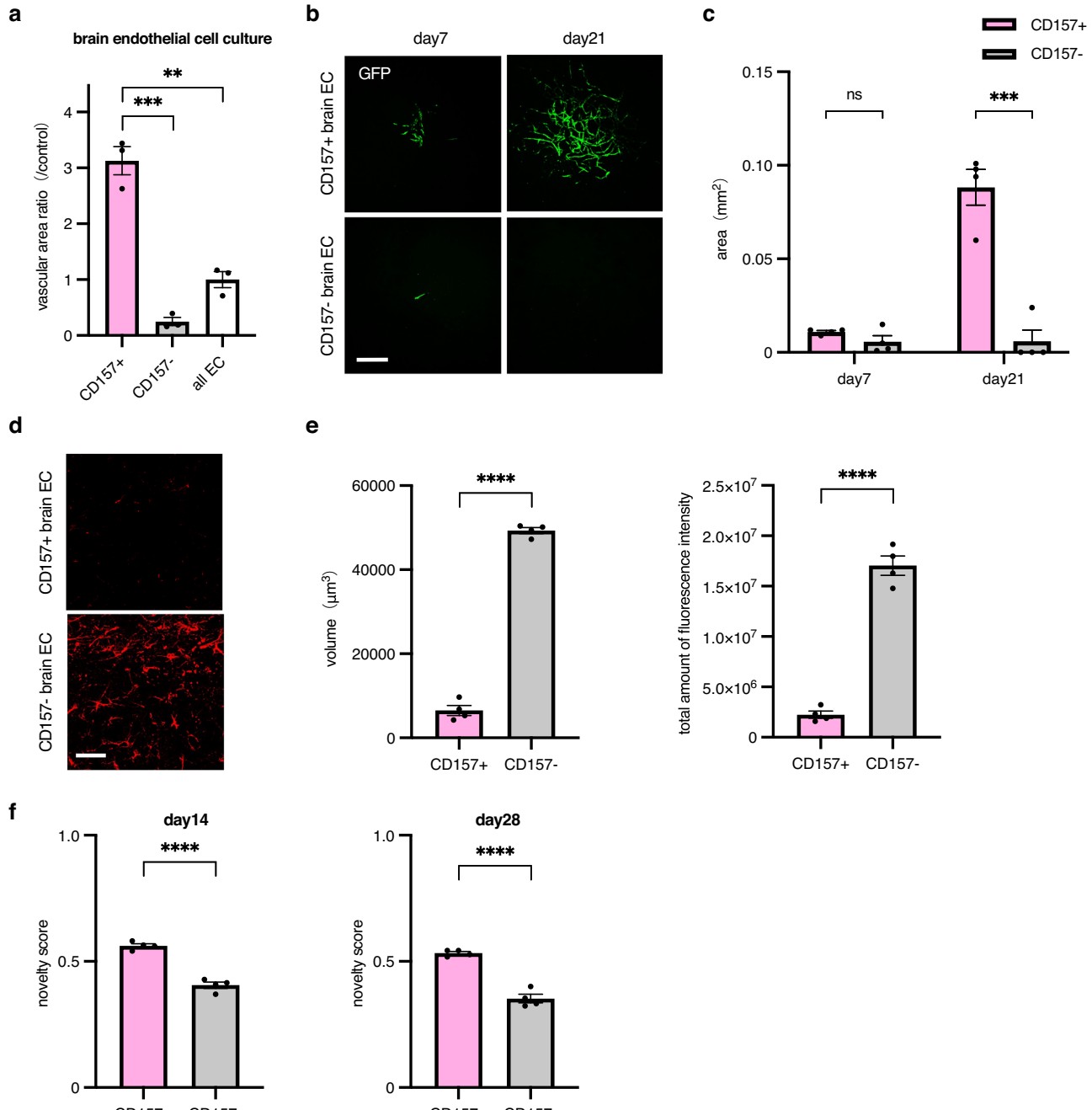

**Fig. 6 | Rescue of fibrosis by CD157-positive ECs. a** Quantification of vascular colony forming–areas by GFP-positive ECs. $5 \times 10^3$ CD157$^+$ or CD157$^-$ ECs from brains were seeded on OP9 feeder cells for 10 days and stained with anti-CD31 antibody (*n* = 3 per group). **b, c** Vascular regeneration by transplanted CD157-positive or -negative ECs from GFP mice into a chronic cerebral hypoperfusion model. Representative images of vascular regeneration observed under TPEM in the same region using the same mice. **b** Vascular regeneration was evaluated on days 7 or 21 after transplantation. The bar graph shows the mean area (days 7, 21) of GFP-positive cells (*n* = 4 per group) (**c**). Data are presented as mean ± SEM. **d, e** Rescue of brain fibrosis by CD157-positive ECs but not by CD157-negative ECs in a chronic cerebral hypoperfusion model. Representative images stained with anti-GFAP pAb (Alexa Flour 546). Scale bar, 50 μm. **e** Bar graphs showing the mean volume and total amount of fluorescence intensity of GFAP-positive cells (*n* = 4 per group). Data are presented as mean ± SEM. **f** Bar graphs showing the mean novelty score on days 14 and 28 (*n* = 4 per group) after the transplantation of ECs. Data are presented as mean ± SEM. **, *P* < 0.01; ***, *P* < 0.001; ****, *P* < 0.0001; ns not significant. EC endothelial cell, GFAP glial fibrillary acidic protein, GFP green fluorescent protein, pAb polyclonal antibody, TPEM two-photon excitation microscopy.

calculated (N/N + F) for intergroups by an independent grader. Exploration behavior was defined as sniffing or touching either object at a distance of 1 cm from the snout. Sitting on the object was not considered exploration. Care was taken to avoid olfactory stimuli by cleaning the apparatus and target objects with 70% alcohol between trials. All sessions were video recorded with a camera positioned above the arena and analyzed by an experimenter blinded to treatment conditions.

The same procedure was utilized to record and calculate the novelty score.

**Quantitative reverse transcription PCR**
Methods for quantitative reverse transcription PCR (RT–qPCR) were previously described in ref. [52]. Briefly, total RNA was extracted from cells using RNeasy-plus mini kits (Qiagen, Hilden, Germany) and reverse

transcribed using a PrimeScript RT regent kit (Takara, Tokyo, Japan) according to the manufacturer's protocol. Real-time PCR analysis was performed using Platinum SYBR Green qPCR SuperMix-UDC (Invitrogen, Carlsbad, CA, USA) and an Mx3000p QPCR System (Agilent). Primers used in these studies were as follows: mouse BDNF 5-GCGGACCCA TGGGACTCT (forward) and 5-CTGCTGCTGTAGTGACCGA (reverse); mouse NGF 5- TCCACATGGGGGAGTTCTCAG (forward) and 5-GCTGAGCACACACACACAGGC (reverse); mouse neurotrophin (NT) 3 5-GGTGAACAAGGTGATGTCCATC (forward) and 5-GCTGCC CACGTAATCCTCCA (reverse); mouse neurotrophin (NT) 4 5′-TCCC TCGCCACTCCTGTTCT-3′ (forward) and 5′-ACTCAGGGGCCAGAAA TGGG-3′ (reverse); mouse EGF 5′-GCCGGCAGATGGGAATGGTT-3′ (forward) and 5′-GTCTTTCTCGCTGGGACCCA-3′ (reverse); mouse FG F2 5′- AGGAAGATGGACGGCTGCTG-3′ (forward) and 5′-GCCCAGTT CGTTTCAGTGCC-3′ (reverse); mouse hepatocyte growth factor (HGF) 5′-ACTCTTGACCCTGACACCCCT-3′ (forward) and 5′-CCCTGTAA CCTTCTCCTTGGCC-3′ (reverse); mouse IGF−1 5′-TGGACCAGA-GACCCTTTGCG-3′ (forward) and 5′-TCCTCAGATCACAGCTCCG GA-3′ (reverse); mouse neuropilin-1 5′-ATCCAAGCTCCGGAACCCTA-3′ (forward) and 5′-CCACAGAACTTCCCCCACAG-3′ (reverse); mouse PDGF-B 5′-CACCTCGCCTGCAAGTGTGA-3′ (forward) and 5′-CGCCTTGTCATGGGTGTGCT-3′ (reverse); mouse VEGF-A 5′-GCCAG CACATAGGAGAGATG-3′ (forward) and 5′-AAATGCTTTCTCCGCT CTGA-3′ (reverse); mouse VEGFR-2 5′-CCCAGCATCTGGAAATCCTA-3′ (forward) and 5′-CCGGTTCCCATCTCTCAGTA-3′ (reverse); mouse Unc 5 A 5′-GCTTCCAGCCTGTCAGCATC-3′ (forward) and 5′-AGAG-CATCGTGGGTGTCGTG-3′ (reverse); mouse Unc 5B 5′-CCTCTCA-GACACGGCCAACT-3′ (forward) and 5′-TGAGTGGGGCTGGATT GGTG-3′ (reverse); mouse Unc 5 C 5′-TAACCTGAAGAAC CAGAGCC-3′ (forward) and 5′-AGGGTCCAGGAGAGGTAAGT-3′ (reverse); mouse Unc 5D 5′-ATTGAGAATGCCAGCGACAT-3′ (for-ward) and 5′-TGTCCACACAGTAAACTCTC-3′ (reverse); mouse down syndrome cell adhesion molecule (DSCAM) 5′-CTGCACTGCCAA-CAACTCGG-3′ (forward) and 5′-CCAATTACGCGGCAGTGGAT-3′ (reverse); mouse deleted in colorectal cancer (DCC) 5′-AACGCT GTCTGTGGACCGAG-3′ (forward) and 5′-GTTGCTTCATTAGCCCT TCC-3′ (reverse); mouse neogenin 5′-TCAGATGATCGACGCCAGCT-3′ (forward) and 5′-GTCCCAGCATCATCCTCAGT-3′ (reverse); mouse glyceraldehyde-3-phosphate dehydrogenase (GAPDH) 5′-TGGCAAAGT GGAGATTGTTGCC-3′ (forward) and 5′-AAGATGGTGATGGGCTTC CCG-3′ (reverse). Results were normalized to GAPDH using a comparative threshold cycle method.

## Statistics and reproducibility
Data are shown as mean ± SEM. Statistical analysis was conducted by one-way ANOVA or unpaired $t$ test using GraphPad Prism 9 statistical software (GraphPad Software, San Diego, CA, USA). A $P$-value of <0.05 was judged as significant.

## Reporting summary
Further information on research design is available in the Nature Portfolio Reporting Summary linked to this article.

## Data availability
The source data behind the graphs in the manuscript were shown in Supplementary Data. The other data generated during and/or analyzed during the current study are available from the corresponding author upon reasonable request.

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

## Acknowledgements

We thank H. Morimoto, Y, Mori and N. Aikawa for excellent technical assistance. This work was supported by the Japan Agency for Medical Research and Development (AMED) under Grant number (JP21gm5010002, 22ck0106727h0001) and the Japan Society for the Promotion of Science (JSPS) Grants-in-Aid for Scientific Research (S) (20H05698).

## Author contributions

Y.M. and F.M. performed the majority of experiments with support from H.N., Y.N., K.M. and H.K. N.T. supervised the project. Y.M., F.M. and N.T. wrote the main manuscript. All authors reviewed the manuscript.

## Competing interests

The authors declare no competing interests.
