## [Peer Review File · Communications Biology]

Reviewers' comments:

Reviewer #1 (Remarks to the Author):

The manuscript by Matsui and colleagues attempted to examine the roles of the brain-derived endothelial cells (ECs) in the regeneration of white matter lesions in a mouse chronic cerebral hypoperfusion model. They showed that brain-derived endothelial ECs were affected by brain-specific netrin-1 and Unc 5B systems. Furthermore, they found that brain CD157-positive ECs were more proliferative and beneficial in a chronic cerebral hypoperfusion model. Their findings indicate the importance of organ-specific interaction of ECs and microenvironment. The manuscript is well written and has ample background to inform the reader. The data is comprehensive and the design is appropriate to propose the methods to cure chronic cerebral hypoperfusion using CD157-positive ECs. While elucidation of the molecular mechanisms underlying the treatment of chronic cerebral hypoperfusion is crucial, some of their conclusions need to be strengthened by the comments below.

Major comments

1. In Figure 1, they chose three organs, brain, liver, and fat, to examine the differences between donor organs. However, these organs may not be enough to conclude that brain is a superior donor organ. Brain, liver, and fat are organs developed from embryonic ectoderm, endoderm and mesoderm. In order to examine whether the ectodermal organs may serve as a good donor, they should include other types of ectodermal organs such as skin.
2. In Figure 2, they showed that brain-derived ECs exhibit the capability of BBB formation as a brain-specific characteristics. In order to show that ECs derived from liver and fat have the organ specific characteristics, they should perform other types of assays.
3. In Figure 4, authors performed co-culture of three types of ECs with fetal brain cells or OP9 cells, and showed that the proliferation of brain-derived ECs was highest in the presence of fetal brain cells and their supernatant. In order to examine the effects of tissue specific factors on the proliferation of organ-specific ECs, they should perform the co-culture experiments using the cells derived from liver and fat.
4. In Figure 5, authors showed that the effects of fetal brain cells on the proliferation of brain-derived ECs were mediated by netrin-1 and Unc 5B systems. In order to examine whether these effects are organ specific, they should perform equivalent assays using OP9 cells.
5. In order to examine whether the netrin-1 and Unc 5B systems confer the brain-derived ECs with organ-specificity, they should over-express Unc 5B in liver and fat-derived ECs and examine the effects of netrin-1 on their proliferation.
6. In Figure 6, authors showed that brain CD157-positive ECs were more proliferative than CD157-negative ECs. They should examine whether the positive effects of netrin-1 on the proliferation of brain-derived ECs are observed only in CD157-positive cells. While they previously reported that these phenomenon were observed in liver-derived, they are encouraged to examine the effects of netrin-1 on the proliferation of CD157-positive liver-derived ECs.

Minor point

In Figures 1, 3, 4, 5 and 6, there are two types of notation for the Y axis of the graphs, "0", "0.0" and "0.00". They should be unified to "0".

Reviewer #2 (Remarks to the Author):

Vascular ECs are an important target for treating chronic cerebral hypoperfusion. The authors demonstrated that brain-derived endothelial cells were more protective against hypoperfusion both in vivo and in vitro models, and clarified the important role of netrin-1 and UNC5B systems in this process. They also found that brain CD157+ ECs were more proliferative and beneficial in the chronic cerebral hypoperfusion model than that in the CD157- ECs upon inoculation. In general, the study is well designed, the methods used are feasible, the results are reliable. The specific critique are as follows:

Major comments:

- 1) Line 81-83: Although liver derived ECs have advantage of stem cell features, why only liver and fat are selected for comparison in the process of specific organ isolation of ECs. Further validation should be conducted on ECs in other organs such as the lungs and heart.
- 2) GFAP is widely used as a marker to activate astrocytes. However, using only GFAP+ cells for statistical fluorescence intensity and volume to represent the degree of white matter damage is not convincing. It is highly recommended to increase more markers such as MBP staining of the myelin sheath to further characterize white matter damage.
- 3) After transplantation of organ specific ECs, in order to detect whether they form functional BBB, authors mentioned that the detection of claudin-5, NG2, and GFAP expression. The molecules mentioned above only represent the composition and position of BBB, which is not sufficient to indicate that the formed BBB is functional. Further identification of the permeability of newly formed BBB is needed through IgG or fluorescence imaging methods, etc.
- 4) Because NG2 is used as a marker of oligodendrocyte progenitor cell, it is inappropriate to represent pericyte coverage. NG2 or CD13 and PDGFR β co-localization is one of the two methods often used to characterize pericyte, which is better illustrate the coverage of pericyte around new blood vessels.
- 5) Compared to ECs derived from liver and fat, ECs derived from brain express high levels of BDNF, FGF, and PDGF-B factors. Why only emphasize BDNF as a candidate for protection here? Please clarify.
- 6) Brain derived ECs exhibit stronger proliferative potential after transplantation. Why there is no difference in the expression of VEGFA and VEGFR-2 among ECs from different organs in Fig. 3b?
- 7) Only new object recognition is not enough for the behavioral evaluation, more test such as the eight-armed maze or water maze experiment are recommended.
- 8) When studying the effects of netrin-1 and UNC 5B on EC proliferation, only in vitro study appears not enough, adding in vivo experiments are recommended.
- 9) When co-cultured the fetal brain cells and ECs together, authors reported that "...similar effects to those observed with the co-culture of ECs with fetal brain cells were observed...". In this case, the role of secreted factors could not be ruled out. ("...It may be that direct cell-cell contact is involved in the alteration of EC proliferation..."). Furthermore, the absence of separate co-culture of neurons and ECs did not indicate that the secreted molecules come from neurons.

10) minor suggestions:

- a) The resolution of all immunofluorescence images needs to be increased to make the images clearer, especially the signal strength of GFAP (such as Fig.1d, 2a, 2b, 2c). Moreover, the lack of positive cell

signals in fluorescence images is not convincing enough. It would be better to display a low magnification image of transplanted endothelial cells.

b) Although the figure legend indicates " $*$, $p < 0.05$ ", there is no significant difference in the identification of " $*$ " in the bar chart.

c) The annotation does not explain the meaning of the vertical axis of the bar graph in the icon, such as how to calculate the growth rate of from day 7-day 21 in Fig. 1c. This type of situation also occurs in other bar chart vertical coordinates, such as "positive rate (%)".

Referee #1: angiogenesis

Referee #2: Netrin - 1 signalling

Reviewers' comments:

Reviewer #1 (Remarks to the Author):

The manuscript by Matsui and colleagues attempted to examine the roles of the brain-derived endothelial cells (ECs) in the regeneration of white matter lesions in a mouse chronic cerebral hypoperfusion model. They showed that brain-derived endothelial ECs were affected by brain-specific netrin-1 and Unc 5B systems. Furthermore, they found that brain CD157-positive ECs were more proliferative and beneficial in a chronic cerebral hypoperfusion model. Their findings indicate the importance of organ-specific interaction of ECs and microenvironment. The manuscript is well written and has ample background to inform the reader. The data is comprehensive and the design is appropriate to propose the methods to cure chronic cerebral hypoperfusion using CD157-positive ECs. While elucidation of the molecular mechanisms underlying the treatment of chronic cerebral hypoperfusion is crucial, some of their conclusions need to be strengthened by the comments below.

Thank you very much for your valuable comments to improve our work. We have amended our paper in accordance with your suggestions.

Major comments

1. In Figure 1, they chose three organs, brain, liver, and fat, to examine the differences between donor organs. However, these organs may not be enough to conclude that brain is a superior donor organ. Brain, liver, and fat are organs developed from embryonic ectoderm, endoderm and mesoderm. In order to examine whether the ectodermal organs may serve as a good donor, they should include other types of ectodermal organs such as skin.

When cultured *in vitro*, the proliferative capacity of primary endothelial cells (ECs) depends on which organ they are derived from. Endothelial cells derived from the heart, skin, or lung did not show higher proliferation compared to those from the liver or fat (Supplementary Fig. 1). Therefore, we determined that ECs from other organs were not suitable for transplantation. We described the reasons for organ selection on lines 85–89.

2. In Figure 2, they showed that brain-derived ECs exhibit the capability of BBB formation as a brain-specific characteristics. In order to show that ECs derived from liver and fat have the organ specific characteristics, they should perform other types of assays.

Thank you for your comment. Your question is very critical to understanding tissue-specific vascular structures formed by organ-specific ECs. We previously observed that liver-derived ECs underwent tissue-specific differentiation (liver sinusoids) in our previous paper on transplantation in the liver. We added this contribution on organ-specific ECs to the discussion section (lines 243–249).

3. In Figure 4, authors performed co-culture of three types of ECs with fetal brain cells or OP9 cells, and showed that the proliferation of brain-derived ECs was highest in the presence of fetal brain cells and their supernatant. In order to examine the effects of tissue specific factors on the proliferation of organ-specific ECs, they should perform the co-culture experiments using the cells derived from liver and fat.

We sorted CD31⁻ CD45⁻ cells (non-endothelial or hematopoietic cell fraction) from fat or liver and co-cultured these with each organ-specific EC line using OP9 feeder cells. The proliferation of vascular ECs differed when cocultured with cells from different organs as partner cells (Supplementary Fig. 6). Considering the precise molecular mechanisms on why such differences existed have not been elucidated, further investigation is required. We added a description of this finding to the manuscript (lines 175–186).

4. In Figure 5, authors showed that the effects of fetal brain cells on the proliferation of brain-derived ECs were mediated by netrin-1 and Unc 5B systems. In order to examine whether these effects are organ specific, they should perform equivalent assays using OP9 cells.

All co-culture experiments were performed in the presence of OP9 feeder cells. We apologize for the confusion. In order to avoid confusing readers, we added "...on OP9 feeder cells" on lines 193–194.

5. In order to examine whether the netrin-1 and Unc 5B systems confer the brain-derived ECs with organ-specificity, they should over-express Unc 5B in liver and fat-derived ECs and examine the effects of netrin-1 on their proliferation.

In accordance with the reviewer's suggestion, ECs derived from the liver were transduced with Unc 5B-expressing lentiviral vectors and the effects of netrin-1 on their proliferation examined. Similar to brain ECs, the proliferation of Unc 5B-expressing liver ECs was enhanced by netrin-1 (Supplementary Fig. 8). We described this finding on lines 208–213. However, in the case of primary fat ECs, these died after lentivirus infection for an unknown reason. Therefore, we show findings for the overexpression of Unc5B in the case of ECs from the liver only.

6. In Figure 6, authors showed that brain CD157-positive ECs were more proliferative than CD157-negative ECs. They should examine whether the positive effects of netrin-1 on the proliferation of brain-derived ECs are observed only in CD157-positive cells. While they previously reported that these phenomenon were observed in liver-derived, they are encouraged to examine the effects of netrin-1 on the proliferation of CD157-positive liver-derived ECs.

As suggested by the reviewer, CD157-positive ECs derived from each organ (brain, liver, and fat) were cultured in the presence of netrin-1. We found that the proliferation of liver- and fat-derived ECs was inhibited, but that of brain-derived ECs was enhanced, by the effect of netrin-1 (Supplementary Fig. 9). We described this on lines 224–226.

Minor point

In Figures 1, 3. 4. 5 and 6, there are two types of notation for the Y axis of the graphs, “0”, “0.0” and “0.00”. They should be unified to “0”.

Thank you for pointing this out. We amended this as suggested.

Reviewer #2 (Remarks to the Author):

Vascular ECs are an important target for treating chronic cerebral hypoperfusion. The authors demonstrated that brain-derived endothelial cells were more protective against hypoperfusion both *in vivo* and *in vitro* models, and clarified the important role of netrin-1 and UNC5B systems in this process. They also found that brain CD157+ ECs were more proliferative and beneficial in the chronic cerebral hypoperfusion model than that in the CD157- ECs upon inoculation. In general, the study is well designed, the methods used are feasible, the results are reliable. The specific critique are as follows:

Thank you very much for your valuable comments to improve our work. We have amended our paper in accordance with your suggestions.

Major comments:

1) Line 81-83: Although liver derived ECs have advantage of stem cell features, why only liver and fat are selected for comparison in the process of specific organ isolation of ECs. Further validation should be conducted on ECs in other organs such as the lungs and heart.

When cultured *in vitro*, the proliferative capacity of primary endothelial cells (ECs) depends on which

organ they are derived from. Endothelial cells derived from the liver, fat, and brain show better proliferative potential than those from other organs, such as the heart, skin, and lung (Supplementary Fig. 1). Therefore, we determined that ECs from the latter organs were not suitable for transplantation. We describe these reasons for organ selection on lines 85–89.

2) GFAP is widely used as a marker to activate astrocytes. However, using only GFAP⁺ cells for statistical fluorescence intensity and volume to represent the degree of white matter damage is not convincing. It is highly recommended to increase more markers such as MBP staining of the myelin sheath to further characterize white matter damage.

As suggested, MBP staining was performed to assess white matter lesions according to the degree of demyelination in lesions. We found that the lowest degree of cerebral ischemia was observed when brain-derived ECs were transplanted (Supplementary Fig. 2). We describe this result on lines 101–104.

3) After transplantation of organ specific ECs, in order to detect whether they form functional BBB, authors mentioned that the detection of claudin-5, NG2, and GFAP expression. The molecules mentioned above only represent the composition and position of BBB, which is not sufficient to indicate that the formed BBB is functional. Further identification of the permeability of newly formed BBB is needed through IgG or fluorescence imaging methods, etc.

Thank you for this comment. We injected TRITC-labeled dextran intravenously into each transplanted mouse and detected leakage from blood vessels. Leakage was not observed from vessels generated by transplanted brain-derived ECs. However, leakage occurred from vessels generated by transplanted fat- or liver-derived ECs (Supplementary Fig. 5). We described this finding on lines 130–133.

4) Because NG2 is used as a marker of oligodendrocyte progenitor cell, it is inappropriate to represent pericyte coverage. NG2 or CD13 and PDGFR β co-localization is one of the two methods often used to characterize pericyte, which is better illustrate the coverage of pericyte around new blood vessels.

As suggested by the reviewer, we performed immunostaining with anti-CD13 antibody. However, unlike for anti-NG2 staining, a strong nonspecific signal was present that seemed to indicate non-pericytes (Supplementary Fig. 4). We describe this finding on lines 122–123. However, if the reviewer recommends deleting this result because it is inappropriate, we will do so.

5) Compared to ECs derived from liver and fat, ECs derived from brain express high levels of BDNF, FGF, and PDGF-B factors. Why only emphasize BDNF as a candidate for protection here? Please

clarify.

As highlighted, it seems that only BDNF was emphasized in our paper. To avoid this, we relabeled the previous Fig. 6g as Supplementary Fig. 10 and rewrote the lines, 231–234.

6) Brain derived ECs exhibit stronger proliferative potential after transplantation. Why there is no difference in the expression of VEGFA and VEGFR-2 among ECs from different organs in Fig. 3b?

It has already been reported that there is no significant difference in VEGF expression between brain and liver ECs (Kalucka *et al.*, 2020, *Cell*). They also reported that VEGFR2 and VEGF signaling-related gene expression is higher in liver than brain ECs. Regarding EC proliferation, it is thought that the effect of cell-cell interactions at the transplant site is greater than the effect of differences in the expression level of VEGF signaling-related genes in transplanted ECs. We have cited this paper on lines 243-249.

7) Only new object recognition is not enough for the behavioral evaluation, more test such as the eight-armed maze or water maze experiment are recommended.

Thank you for this suggestion. We performed a Y-maze test to assess cognitive function. Mice that were transplanted with brain-derived ECs showed better function (Supplementary Fig. 3). We described this result on lines 108–110.

8) When studying the effects of netrin-1 and UNC 5B on EC proliferation, only in vitro study appears not enough, adding in vivo experiments are recommended.

Thank you for this suggestion. Endothelial cells derived from each organ were mixed with netrin-1 in Matrigel and injected subcutaneously into mice in a Matrigel plug assay. In Matrigel plugs mixed with brain-derived ECs, netrin-1-induced EC proliferation was observed (Supplementary Fig. 7). However, ECs from the liver or fat did not proliferate with the addition of netrin-1. We describe this result on lines 196-200.

9) When co-cultured the fetal brain cells and ECs together, authors reported that "...similar effects to those observed with the co-culture of ECs with fetal brain cells were observed...". In this case, the role of secreted factors could not be ruled out. ("...It may be that direct cell-cell contact is involved in the alteration of EC proliferation..."). Furthermore, the absence of separate co-culture of neurons and ECs did not indicate that the secreted molecules come from neurons.

We apologize for this mistake. We rephrased the sentence (line 171).

10) minor suggestions:

a) The resolution of all immunofluorescence images needs to be increased to make the images clearer, especially the signal strength of GFAP (such as Fig. 1d, 2a, 2b, 2c). Moreover, the lack of positive cell signals in fluorescence images is not convincing enough. It would be better to display a low magnification image of transplanted endothelial cells.

Thank you for this advice. As suggested, it seems to be difficult to distinguish images due to fluorescence signals. Therefore, we adjusted immunostaining images to make these as clear as possible.

b) Although the figure legend indicates "**, $p < 0.05$ ", there is no significant difference in the identification of "*" in the bar chart.

Thank you for pointing this out. We checked our manuscript but we could not find the relevant figure legend. Could this please be pointed out?

c) The annotation does not explain the meaning of the vertical axis of the bar graph in the icon, such as how to calculate the growth rate of from day 7-day 21 in Fig. 1c. This type of situation also occurs in other bar chart vertical coordinates, such as "positive rate (%)".

Thank you for pointing this out. We inserted an explanation in Figures 1c and 2 legends.

REVIEWERS' COMMENTS:

Reviewer #1 (Remarks to the Author):

The authors have appropriately addressed the comments raised by the reviewers.

Reviewer #2 (Remarks to the Author):

no more questions. Thanks